# Hakkō Ichiu: Religious Rhetoric in Imperial Japan

## Ziming Wang

Department of Religious Studies, School of Philosophy, Fudan University, Shanghai 200433, China;
wangzm0214@outlook.com

**Abstract:** The wartime propaganda slogan Hakkō Ichiu 八紘一宇 ("Unify the whole world under one roof") was loaded with historical meaning: Japan was glorifying the aggression and colonization of war by fostering a specific interpretation of the narrative about how Jimmu, the first emperor, founded the nation in State Shinto mythology. In this article, I consider this slogan as central to a religious rhetoric with nationalistic overtones and I analyze it in terms of etymology, connotation, and rhetorical devices. First, the expression Hakkō Ichiu originated in ancient East Asian cosmology, before becoming one of the rhetorical expressions of State Shinto, emphasizing the extent of the imperial reign. Second, the Nichirenist activist Tanaka Chigaku rediscovered it and gave it an expansionist connotation, fostering a syncretistic approach mixing Buddhist and Shinto features. Finally, during wartime, in official documents, lyrics, trademarks, etc., the slogan gave way to a number of graphic and monumental expressions, reinforcing its connections with militarism and ultranationalism. The most notable of these material expressions was the Hakkō Ichiu Tower, erected to commemorate the 2600th anniversary of the foundation of the nation and perpetuate the State Shinto rhetoric.

**Keywords:** Hakkō Ichiu; State Shinto; Emperor Jimmu; Kokutai; Nichirenism; religious rhetoric; political mythology

## 1. Introduction

After the outbreak of the Second Sino-Japanese war in 1937, people living in Imperial Japan would frequently notice the four-character formula propaganda slogan *Hakkō Ichiu* 八紘一宇, literally "eight cords with one roof", the meaning of which was: "Unify the whole world under one roof".[1] It is generally believed that this elusive expression was a State Shinto term, derived from the declaration attributed to Emperor Jimmu, the legendary first emperor of Japan and the descendant of the sun goddess Amaterasu, when he proclaimed his accession to the throne. This declaration is recorded in the Japanese Chronicle *Nihongi* 『日本書紀』—the official history of Japan at that time—completed in 720. After the Meiji Restoration, State Shinto was gradually established as the official ideology of Imperial Japan by the government, whose religious accounts would regard the origin of Japanese mythology as authentic history. Therefore, Jimmu's legendary accession became a significant token of the "unbroken Imperial line" 万世一系 that originated from Amaterasu.

From Murakami Shigeyoshi's influential book, *Kokka Shintō*, onwards, scholars have developed successive discourses around the idea of State Shinto and its influence on postwar Japan. Murakami regarded State Shinto as a state religion, which led the country into militarism, ultranationalism, and disastrous wars, by ruling the spiritual world of Japanese people (Murakami 1970). Shimazono Susumu emphasizes State Shinto's influence on education and regards the imperial rites as an important part of State Shinto, which has not been fundamentally abolished to date (Shimazono 2010, 2021). Inoue Hiroshi regards State Shinto as modern Japan's system of official religion in a form that emerged from the theoretical and institutional reorganization of shrines and "Shinto".[2] Some western researchers prefer to treat State Shinto as an "invented tradition". Helen Hardacre analyzes how Shinto

formed new relations with the government through state management, which deeply affected shrines, the priesthood, and shrine communities (Hardacre 1989, 2017). Trent E. Maxey expounded how the Meiji government strove to draw a clear line of demarcation between secular and religious dominations (Maxey 2014). In recent years, scholarship has gradually focused on the relationship between *Kokutai* 国体 ("national polity") and State Shinto, a dimension that had been overlooked in past historical studies. In particular, in 2019, Fudita Hiromasa compiled the results of a symposium, *State Shinto and Kokutai: A study of the interdisciplinarity of Religion and Nationalism*, advancing research in the field (Fudita 2019).

Still, in the studies around the field, a point of importance has been overlooked till now: before being used by State Shinto, the expression *Hakkō Ichiu* had been coined in 1918 by a Buddhist, namely, the Nichiren Buddhist 日蓮宗 scholar, Tanaka Chigaku 田中智学 (1861–1939). Tanaka was aiming at creating a religious philosophy that blended Nichiren beliefs with Japanese nationalism. He launched a movement called Nichirenism 日蓮主義, which has been a hot topic in Japanese Buddhist studies, with Nishiyama Shigeru's discussion on Nichirenism and nationalism and Ōtani Eiichi's recent efforts to divide Nichirenists into three generations (Nishiyama 2016; Ōtani 2001, 2019).

In the late 1930s, following the Marco Polo Bridge Incident, Japan accelerated the process of aggression and expansion, and the term *Hakkō Ichiu* was borrowed as a propaganda slogan. As a rallying cry and a legitimate endorsement of aggression and expansion, around 1940, *Hakkō Ichiu* was widely circulated in newspapers, newsreels, radio, lyrics, stamps, and everywhere, becoming synonymous with Imperial Japanese expansionism. The year 1940 was an unusual year for the Empire of Japan. According to the Japanese Chronicle, historians with official backgrounds speculated that Emperor Jimmu's enthronement took place in 660BC; hence, 1940 was considered the 2600th anniversary of the Empire of Japan. The Japanese held grandiose national commemorations to celebrate the anniversary, the account of which provides us with a window to examine Imperial Japan at its zenith (Ruoff 2010, p. 1). A magnificent monument was erected in Miyazaki 宮崎, a prefecture anciently called Hyūga 日向 in southeastern Kyushu, where (so the mythical narrative tells us) Emperor Jimmu started an eastward expedition and eventually reached Yamato 大和. A colossal *Hakkō Ichiu* four-character inscription was carved on the front tablet, so that the building was named "Hakkō Ichiu Tower".

Not much scholarly research has been undertaken on *Hakkō Ichiu* and its appendants. Walter Edwards has researched the Hakkō Ichiu Tower in Miyazaki as a representation of Japanese wartime ideology, examining how Imperial Japan performatively glorified its expansionism (Edwards 2003, pp. 298–324). A left-wing civic group in Miyazaki, "Hakkō Ichiu Tower Research Association(「八紘一宇」の塔を考える会)," contributed a lot to the research in this field, compiling a detailed account of the construction of the tower and the origin of each denoted stone used in its construction, based on searching for evidence of Japanese war crimes, holding several exhibitions in China (Hakkō 2017). Shimakawa Masashi researched how the propagation of the idea of *Hakkō Ichiu*, together with the Japanization movement in Manchuria, resulted in the construction of a shrine in Manchuria (Shimakawa 1984). Konno Nobuyuki placed the discourse of *Hakkō Ichiu* in the context of State Shinto and examined the debate in Shinto circles over the scope of application of *Hakkō Ichiu* and other issues (Konno 2019, pp. 421–44). Recently, a systematic study of *Hakkō Ichiu* has been published: Kuroiwa Akihiko examines the formula from the perspective of social and intellectual history during wartime and postwar and discovers that, in addition to being utilized by the central institutions of Japan, *Hakkō Ichiu* also had its folk and local expressions (Kuroiwa 2022).

In this article, I analyze the formula *Hakkō Ichiu* as an example of religious rhetoric. First, I examine the spatial connotations of the term from the perspective of its etymology, and how it entered into the discourse of State Shinto. Second, I connect the rediscovery of the expression *Hakkō Ichiu* in modern times to Nichirenism, investigating the way Buddhism gave it a new meaning and, by doing so, triggered a specific kind of "interreligious

dialogue". Finally, I explore the path through which the expression came to represent Japanese wartime nationalist ideology, which leads me to detail the role that the mythology proper to State Shinto played in wartime nationalism and war propaganda. By doing so, I hope to show how specific metaphors become part of an overall narrative that is inextricably religious and political.

## 2. From *Bahong* to *Hakkō*: An Etymology Investigation

The formula *Hakkō Ichiu* can be divided into two parts, *Hakkō* 八紘, and *Ichiu* 一宇. Literally, *Hakkō* means "eight cords", while *Ichiu* translates as "one roof". Etymologically, *Hakkō* 八紘 is derived from ancient Chinese classics, and is pronounced *Bahong* in Standard Mandarin.

### 2.1. Chinese Sources

As recorded in Chapter *Benming* 《本命》 of *Da Dai Liji* 《大戴禮記》, the number Eight (*Ba* 八) was formed by the coincidence of the cosmic principles *Wei* 維 and *Gang* 綱, from which Heaven and Earth were shaped and the union of *Yin* and *Yang* was understood by the Ancient Sages (see Fang 2008, p. 1287).[3] The cosmology behind such an assertion is not entirely clear, as both *Wei* and *Gang* actually have "cord" as their original meaning. In the Qing Dynasty, the scholar Kong Guangsen 孔廣森 (1753–1786) annotated this sentence, stating that the four corners of the cosmos were called *Wei*, and the four directions were called *Gang*.[4] Therefore, *Ba* refers to the eight divisions through which space is structured, and by extension, to cosmic organization. Similarly, *Hong* 紘 also represents cords or strings, and it is sometimes thought to be more or less synonymous to *Wei*.[5] The connection between cords and spatial orientation may be derived from very ancient Chinese cosmological beliefs according to which Heaven is dome-like, supported by nine mountainous pillars, and Earth is square, stretched by four gigantic cords. Thus, *Ba* and *Hong* combined together into one word, designating the world seen from the viewpoint of its spatial organization.

The word *Bahong* first appeared in the Daoist Classic *Liezi* 《列子》, attributed to Lie Yukou 列御寇, the forerunner of the Daoist school in the Spring and Autumn Period in China, though the dating of this work remains very disputed. In any case, the *Liezi* is influenced by the cosmology that develops during the Warring States period. In the chapter *Tangwen* 《汤问》, the *Liezi* presents the reader with a fictional dialogue between Tang of Yin 殷汤 and a sage Xia Ge 夏革 in which the two interlocutors discuss whether the universe is finite or infinite.

> Once, Tang of Yin asked: "Is there a difference between gigantic and wee? Is there a difference between long and short? What are the differences and similarities?"
>
> Xia Ge answered: "There is a vast ocean in the east of the Bohai Sea, I do not know over how many hundreds of millions of miles it extends. It is a bottomless valley. It has no bottom and is named *Guixu* 归墟. All the waters from *Bahong* 八纮 and *Jiuye* 九野, all the waves of the Milky Way, pour here. But its water level is neither increasing nor decreasing". (Yang 1997, pp. 147–51)

This dialogue reveals that the *Liezi* used hyperbole to depict the vastness between Heaven and Earth. Additionally, the expression *Bahong* is used to describe the farthest places on the Earth.

In *Huainanzi* 《淮南子》, an encyclopedic work of the Han Dynasty, Liu An 刘安 (179BC–122BC) and his retainers proposed a more detailed explanation of the word *Bahong* as a terrestrial term. In the chapter "Terrestrial Forms" (*Dixing Xun* 《墜形訓》), the book records: "The borders of each of the nine provinces [*Jiuzhou* 九州] encompass one thousand li. Beyond the nine provinces are eight distant regions [*Bayin* 八殥], each encompassing a thousand li [ ... ] Beyond the eight distant regions are eight outlying regions [*Bahong* 八紘], each also encompassing one thousand li (He 1998, pp. 364–65)". After this, the chapter enumerates the specific names and locations of the *Bahong*. Beyond *Bahong* lies

the end of the Earth, the eight "ultimate regions" [*Baji* 八極]. In contrast to the imaginary world by *Baji*, *Bahong* in the *Huainanzi* still refers to concrete, identifiable regions. However, *Bahong* gradually became an abstract term, more or less a synonym of *Tianxia* 天下, the classical expression for designating "the world" in Chinese culture.

### 2.2. The Japanese Acculturation

After communications between China and Yamataikoku started to occur in the 3rd century, many Chinese cultural elements were introduced to Japan, the more so as exchanges became increasingly intensive. It is not clear when the word *Bahong* was introduced into Japan. However, the term is used in the *Nihongi* (completed, as we already said, in 720AD). As already indicated, *Bahong* is pronounced *Hakkō* in Japanese.

The myth we have summarized in the introduction deserves a closer investigation: Emperor Jimmu, a direct descendant of Amaterasu, led his clansmen on an eastward expedition from Hyūga 日向, an ancient province in the present Miyazaki Prefecture, southeast of Kyushu, using forces to subjugate many ethnic groups, such as the *Emishi* 蝦夷, until he invincibly reached the center of Yamato Province.

After his enemies were driven out, Emperor Jimmu declared in an edict that he would ascend the throne in Kashihara, a place chosen to be his capital where he would build a palace in order to manifest the glory of his ancestors. In the edict, Emperor Jimmu asserted: "Thereafter the capital may be extended so as to embrace all the six cardinal points, and the eight cords may be covered so as to form a roof. Will this not be well?" (Translation by Aston 1972, p. 131). This is the original provenance of the expression *Hakkō Ichiu* in Japan, and this edict was called *Kashihara Kentō no Rei—Hakkō Nariu no Mikotonori* 「橿原建都の令一 八紘為宇の詔」, i.e., "The decree of capital founding in Kashihara—The imperial edict of *Hakkō Nariu*" by later historians (Kuroiwa 2022, p. 161). Emperor Jimmu became the founding ancestor of the imperial lineage, advocating Japan's perpetual rule according to the imperial chronicles and State Shinto mythology.

Apparently, in Jimmu's edict, *Hakkō* takes the figurative sense of *Tianxia* ("the world"), less ambiguous than the one it takes when the formula resurged in the early 20th century as a token of Japanese expansionism. Jimmu unquestionably referred to the world, albeit he had no idea of its actual expanse. In contrast, it was clear that the imperial army and navy were unable to rule all over the world in the Pacific War, thereby making the boundaries of expansion represented by *Hakkō* become a contentious subject. Moreover, the single character 宇 means "roof". Its meaning was extended to the house in its entirety, coming to refer to the family or household, with an affective connotation.

With its establishment in 1868, the new Meiji government was eager to establish the new regime's legitimacy. Although the Meiji government, before the creation of parliament in 1890, was, de facto, an oligarchy, it was ostensibly an absolute monarchy ruled by the emperor. In consideration of the elements of veneration of the emperor and the idea of *Kokutai* from *Fukkō Shintō* 復古神道, advocated by Hirata Atsutane 平田篤胤 in the Edo period (Hardacre 2017, p. 348), the government spontaneously took note of Shinto. On March 13th, the new government, acting in the name of the young emperor, proclaimed a decree that the new country would: "*Ōsei Fukko* 王政復古 (restore the antique imperial rule)"; "*Saisei Itchi* 祭政一致 ([ensure the] unity of ritual and government)"; and also reestablish the *Jingikan* 神祇官, i.e., the "Department of Divinities", an imperial bureau first established in the 8th century in order to administrate rituals and emperors' mausoleums (Yasumaru and Miyachi 1988, p. 425). The next day, March 14th, the emperor promulgated the famous Charter Oath as a fundamental document of the Meiji Restoration, setting up Japan's modernization. Thus, by adopting a series of measures, including the so-called *Shinbutsu Bunri* 神仏分離 (the separation of Shinto from Buddhism), the government gradually established the regime of State Shinto, an official "quasi-religion"[6].

The March 13th Decree stipulated that the government would restore imperial rule based upon the foundations established by Emperor Jimmu. The expression "Promoting the Jimmu's spirit" appeared in many other documents.[7] Based on the Shinto ritual, the

government formulated a comprehensive ritual calendar, marked by a number of ceremonies. There were two festivals and imperial rituals related to Emperor Jimmu, *Jimmu Tennōsai* 神武天皇祭, and *Kigensetsu* 紀元節. Established in 1871, *Jimmu Tennōsai* was meant to commemorate the anniversary of the demise of Emperor Jimmu on April 3rd, and *Kigensetsu* (the commemoration of the enthronement of Jimmu), celebrated on February 11th, was established in 1873. The emperor was required to participate personally in the rituals of both festivals (Hardacre 2017, pp. 363–64). *Kigensetsu* was listed as one of the most important festivals of Imperial Japan before being abolished in 1948 by the General Headquarters (GHQ). Nevertheless, it resurged in 1967 by changing its name to "National Foundation Day". In the State Shinto narrative, Emperor Jimmu, as the first earthly emperor, could be ranked only second to Amaterasu, and his eastward expedition was later regarded as typological of Japanese colonial expansion. However, the expression *Hakkō Ichiu*, as relating to Emperor Jimmu, rarely appeared throughout the Meiji era. A more common expression was *Tenjō Mukyū* 天壤無窮 ("eternal as Heaven and Earth"), also derived from the *Nihongi*, aiming to describe the successive reigns composing the imperial line as one perpetual reign.[8]

*2.3. The "National Polity" in Time and Space*

At this point, a related concept needs to be introduced into our analysis. *Kokutai* is commonly translated as "national polity" and can be said to designate the national structure. *Kokutai* was at the basis of the emperor's sovereignty and served as the official ideology of the Japanese Empire, part of which, including the core content of *Tennō Sūkei* 天皇崇敬 (emperor reverence), overlapped with some ideas of State Shinto but also incorporated some Confucianism. It was elaborated in the *Imperial Rescript on Education*, proclaimed in 1890, and distributed to secondary and primary schools with a portrait of the emperor shortly afterward. Simultaneously, students were required to recite and revere it. In the beginning of the *Imperial Rescript on Education*, one reads:

> "Our Imperial Ancestors have founded Our Empire on a basis broad and everlasting and have deeply and firmly implanted virtue; Our subjects ever united in loyalty and filial piety have from generation to generation illustrated the beauty thereof. This is the glory of the fundamental character of *Kokutai*, and herein lies the source of Our education." (Monbushō 1909, p. 8)

In addition to emphasizing the continuous imperial line, the stress on the subjects' loyalty to the emperor was also an essential part of *Kokutai*. Walter Edwards investigated the phrase *Tenjō Mukyū* in a work by the Meiji era historian Kita Sadakichi 喜田貞吉 *Kokushi no Kyōiku* 『国史之教育』, a reference book for history teachers published by the Ministry of Education in 1910 (Edwards 2003, pp. 301–2). In this book, Kita repeatedly discussed the conception of *Tenjō Mukyū* and proposed that:

> "Loyalty and filial piety are absolute obligations to our emperor and father. We are subjects of a continuous imperial line that remained unchanged for over 2500 years. The relationship between the emperor and subjects is *Tenjō Mukyū*, which is as real as the parent–child relationship. How fortunate that we are born in such *Kokutai*!". (Kita 1910, p. 62)

The *Imperial Rescript on Education* was not a document limited to the field of education but was also a canon that sanctified *Kokutai* and controlled all of society. The rescript itself also became sacrosanct, promoting the establishment of a patriarchal imperial state.

Kita also wrote that the *Tenjō Mukyū* relationship between the emperor and his subjects was precisely what made Japan superior to other countries (Kita 1910, pp. 1–2, 30). Still, the intent of the *Tenjō Mukyū* relationship did not have an expansionist meaning. Kita asserted that the relationship was only bestowed to the subjects in the Japanese archipelago, and foreign territories were excluded (Kita 1910, p. 104). The *Tenjō Mukyū* catchphrase was meant to emphasize the immutability of the unbroken imperial line in *time*, while the *Hakkō Ichiu* expression was emphasizing the *spatial scope* of imperial domination.

### 3. Emperor Jimmu's Spirit and Nichirenism

Although the mythology of Emperor Jimmu was enjoying a critically important status in the narrative of State Shinto, the formula *Hakkō Ichiu* did not appear until the 1910s, and the word was coined by the Nichiren Buddhist preacher and scholar Tanaka Chigaku 田中智学. As the third son of a noted physician in a Nichirenist family, he was born in 1861, the same year when the famous Japanese Christian evangelist Uchimura Kanzō 内村鑑三 was born. Ōtani Eiichi 大谷栄一, a specialist in Japanese Buddhism, has drawn a striking parallel: "If Uchimura Kanzō loved two J (Jesus and Japan), it could be said that Tanaka Chigaku devoted his life to two N (Nichiren and Nippon)." (Ōtani 2019, p. 45).

*3.1. Tanaka Chigaku and the Propagation of Nichirenism*

Tanaka was born in Edo and was first given the name of Tomoenosuke 巴之助. When he was ten years old, Tomoenosuke joined the Nichiren sect as a monk, and received the Dharma name Chigaku 智学, meaning "Wisdom and learning". After seven years as a monk, he resumed secular life, becoming a lay preacher of the Nichiren sect and initiating lay Buddhist campaigns. In 1884, he established his Nichiren study group, *Risshō Ankokukai* 立正安国会, composed entirely of laypeople. The name of the group was derived from the treatise *Risshō Ankoku Ron* 『立正安国論』 of Nichiren: "To assert the doctrine of Nichiren as the basis for state construction". The group changed its name to *Kokuchūkai* 国柱会 ("Pillar of the Nation") in 1914, a name that was derived from Nichiren's words, "I will be the pillar of Japan （我日本の柱とならん）".

Tanaka believed, in line with Nichiren Buddhist teachings, that the *Lotus Sūtra* recorded the highest and the most genuine Dharma and that other sūtras and other sects were not transmitting the authentic Dharma (Ōtani 2019, p. 50). Additionally, he defined his thoughts as Nichirenism[9] to distinguish from other sects that have faith in Nichiren, where he wrote that: "In terms of religion, it is called the Nichiren sect, and when it comes to the sūtra, it is also called the 'Lotus sect.' However, when used in a broader sense than pure faith, ideology, or even life awareness, [this doctrine] generally should be called Nichirenism (Tanaka 1936, p. 17)".

In 1910, Tanaka began to promote reform of the Nichiren sect and wrote a reform manifesto entitled *Shūmon no Ishin* 『宗門之維新』. In this work, he implies an expansionist and even cosmopolitan view, asserting that "the doctrine of Nichiren is not just for our sect, but for the whole state. That is to say, Nichiren's tenets should protect Japan, and in the future, humanity all over the world must have a common destination (faith in Nichiren), which is the highest Dharma of karma (Tanaka 1901a, p. 2)". In the appendix of this book, he gave a timetable *Myōshō Mirai Nenhyō* 『妙宗未来年表』 for the development of Nichirenism over the next 50 years, dividing it into ten periods. He projected that the Imperial Diet would establish Nichiren Buddhism as the state religion, and other religions in the country would be dismissed. Afterward, becoming the state religion, the Nichiren Sect would set up a network of ordination platforms in the country (*Kokuristu Kaidan* 国立戒壇), and dispatch missionary groups abroad, also establishing overseas mission headquarters to preach Dharma, and expecting the unification of the world through Nichiren Buddhism (Tanaka 1901b, pp. 8–28). Seemingly, Tanaka constructed a Utopian blueprint that was taking its model from the expansion of Catholicism in the Age of Discovery.

After two victories in the First Sino-Japanese War and the Russo-Japanese War, nationalism and national pride peaked in Japan. The government took a series of actions, such as propagating the *Kokutai* mythology and doctrine in schools. Quite naturally, Tanaka endeavored to engineer an amalgamation of Nichiren Buddhism and *Kokutai* ideology. Based on Nichiren doctrines, he proposed the conception of *Hōkoku Myōgō* 法国冥合, meaning the integration of Buddhism and politics, dividing it into three stages that can be summarized as follows: First, the state should comprehensively implement *Kokutai* and popularize Buddhism throughout the whole country. Second, the emperor needed to issue an edict to establish altars and, at the same time, the Diet was urged to amend Article 28 of the Constitution on the freedom of religious belief so as to establish Nichiren Buddhism as the State

Religion, thus achieving the integration of religion and politics. Finally, according to the way the Dharma was expounded in the *Lotus Sūtra*, the unification of thought, religion, morality, and politics would be realized in the world (Ōtani 2019, pp. 140–41).

### 3.2. *The Emperor, the Nation and the World*

Tanaka flexibly utilized the syncretic tradition of Japanese religions, remolding the *Kokutai* by adding some Buddhist elements, as he transformed the emperor into the *cakravartin*, an ideal universal ruler in Indian mythology. The imperial family was to be the highest flamen of Nichiren ordination platforms. Tanaka claimed that the imperial seal of Japan was a 16-petalled chrysanthemum, which originated from the *dharmacakra* of *cakravartin* in Indian thought, a result of his belief in the fact that there had been a royal family meant to unify the world from time immemorial, and that this pattern has been passed down to the present (Tanaka 1910, p. 17). The presence of the emperor was indispensable to Nichirenism, Tanaka asserted.

In 1903, Tanaka's study group organized a journey to Emperor Jimmu's mausoleum. Tanaka gave a four and a half hours' speech that was later compiled and presented to the soldiers who went to the Russo-Japanese War the following year in an essay called *Sekai Tōistu no Tengyō* 「世界統一の天業」 ("The Sacred Work of World Unification"), which expounded Tanaka's theory of *Kokutai*. Obviously, his *Kokutai* theory was also centered on the emperor and imperial reign line, and Emperor Jimmu became a fixture of his discourse. In his exposition, Emperor Jimmu's enterprise of the national foundation represented two spirits, one called *Jūki* 重暉 and another called *Yōsei* 養正. *Jūki* was about cherishing the virtues accumulated for a long time in the past. It was the virtue of the Heaven, and, as the sun is the most prominent of all celestial signs, such spirit found an ideal representation in the sun, which is why the country was called "sun's origin". *Yōsei* meant the design of practicing virtue forever in the future. It represented the virtue of the Earth, and the most prominent object on the Earth is the mountain, used to express the virtue of longevity, and this was why the country was called "Yamato,"[10] the Fuji being a national landmark (Tanaka 1910, pp. 10–11). Emperor Jimmu not only founded the nation but also embodied the virtues of Heaven and Earth, which means that he also endeavored to unify the world through the practice of virtue.

In 1913, Tanaka published an essay called *Emperor Jimmu's National Foundation* on *Kokuchū Shinbun* 『国柱新聞』, the official newspaper of *Risshō Ankokukai*. He selected the paragraph from the *Nihongi* where the expression *Hakkō Ichiu* appears, for expressing the ambition that world unification will be realized under the auspices of Emperor Jimmu. The formula *Hakkō Ichiu* subsequently started to circulate. In fact, what Tanaka wanted to emphasize was the sentence preceding the expression *Hakkō Ichiu*, in which Emperor Jimmu says: "Above, I should then respond to the kindness of the Heavenly Powers in granting me the Kingdom, and below, I should extend the line of the Imperial descendants and foster rightmindedness (Translation by Aston 1972, p. 131)". Tanaka's focus is on the word *Yōsei*, translated here as "to foster rightmindedness", read as being Jimmu's manifesto for world unification and as setting the historical mission of the Japanese people (Tanaka 1922, p. 277).

For the first time in Imperial Japan, Tanaka had rediscovered the term *Hakkō Ichiu* from the *Nihongi* and had given it a connotation of expansionism and even of world unification, all of which was undoubtedly based on his innovative Nichirenist theory of *Kokutai*. As Yulia Burenina commented, "Nichirenism seems to have enabled the coexistence of Japan-centralism and cosmopolitanism (Burenina 2020, p. 219)". Burenina believes that Tanaka attempted to reconsider Japanese uniqueness as embodying a universal truth with a supernational perspective (Burenina 2020, p. 219).

It is noteworthy that, following Nichiren's notion of an "unprecedented great struggle" as expressed in the *Senjishō* 撰時抄, Tanaka predicted that there would be a great world war just before the world's unification, although Tanaka himself preferred to unify the world through moral means (Ōtani 2006, pp. 89–92). This theory was inherited and put

into practice by Ishiwara Kanji 石原莞爾, a member of the *Kokuchūkai* and the disciple of Tanaka, who later became world-famous for plotting the Manchurian Incident.[11] Ironically, the phrase *Hakkō Ichiu*, coined from Nichirenism, eventually became a propaganda slogan for starting a World War but ended up merely symbolizing the utopian dream nurtured by Imperial Japan.

## 4. *Hakkō Ichiu* during Wartime

With the unexpected outbreak of the Marco Polo Bridge Incident in the summer of 1937, Imperial Japan rapidly became embroiled in a conflict with China while beginning to accelerate its aggression and colonization in East and Southeast Asia.[12] The Japanese government and Imperial Japanese Armed Forces soon found themselves confronted with two problems: one was how to conduct a national mobilization, and the second was how to establish legitimacy for the war.

### 4.1. From Hakkō Ichiu to "Greater East Asia"

In response to the first problem, the government launched the National Spiritual Mobilization Movement following the emperor's edict proclaimed in September 1937, calling on soldiers to fight bravely and the nation to overcome difficulties with loyalty and perseverance (Seo 1939, pp. 1–8). In November, the Ministry of Education released a pamphlet entitled "The Spirit of *Hakkō Ichiu*" to promote patriotism and loyalty in the context of the National Spiritual Mobilization Movement. The pamphlet's contents were mainly based on the State Shinto creation myth and the "justification" for the Second Sino-Japanese War. Still, it specifically explained what *Hakkō Ichiu* meant, i.e., to create a unified and orderly community of harmony ruled by the emperor, said otherwise: "To make people become subjects of the emperor" (*Kōka* 皇化). The pamphlet also proclaimed that "*Hakkō Ichiu* means that all disasters are removed through *Kōka*, not only in Japan but also in every country and every nation, so that each country and nation can not only stand on its own feet but also support each other as a harmonious family". In addition, it emphasized that *Hakkō Ichiu* did not mean the same thing as having a foreign hegemonic state annexing the territory of another country: "It was the ultimate goal of our subjects to support imperial reign within the brotherhood" (Monbushō 1937, pp. 10–11). The Japanese government intended to cloak its aggression in a veneer of familial harmony to distinguish it from the violent colonial expansion of the European powers, thereby concealing its expansionist intentions.

To tackle the second problem, the government harnessed the ideological resources of State Shinto and *Kokutai*. With Japan mired in the Sino-Japanese War, Prime Minister Konoe Fumimaro 近衛文麿 issued his second declaration on 3 November 1938, changing the previous stress on "self-defense" to one on "Establishing a New Order in East Asia" (which later on was changed for the notorious slogan "Greater East Asia Co-Prosperity Sphere") as a way to justify and glorify the war of aggression being led. In the pronouncement, Konoe mentioned that "the establishment of a new order in East Asia is in complete conformity with the very spirit in which the Empire was founded; to achieve such a task is the exalted responsibility with which our present generation is entrusted (Konoe 1938)". The mention of the "spirit of Jimmu" reappeared in official documents, tied to the current mission of Japan.

On 22 July 1940, Konoe established his second cabinet, and, four days later, the government proposed its "Basic Guideline for the Nation Policies (Yagami 2006, pp. 88–89)". The first article of this document stated that "the national policy of the imperial state shall lead to the establishment of world peace based on the great spirit of the national foundation with *Hakkō Ichiu* [as its basis]. The first principle bears on constructing a new order in Greater East Asia, through the firm bond of Japan, Manchuria, and China, with the imperial state as its core (Konoe 1940)". While Tanaka Chigaku envisioned that *Hakkō Ichiu* would include the whole world, so as to build a universal brotherhood, the reference to "Greater East Asia" found in Konoe's proclamation was extremely vague, but it did imply

that *Hakkō Ichiu* had territorial boundaries. Especially after the signing of the Tripartite Pact in September 1940, Japan, Germany, and Italy mutually recognized each other's scope and order within their respective spheres.

Nonetheless, there were still different voices in the Diet querying the blueprint of the "New Order in East Asia". On 2 February 1940, Diet member Saitō Takao 斎藤隆夫 made a notable speech to the Diet, questioning the so-called "New Order in East Asia" as just a pretext to justify the war with China, which the military was constantly involved in but could not finish. He argued, "The ideals of Chinese *Ōdō* 王道 (benevolent rule) and *Hakkō Ichiu* are difficult to understand for those of us who are actually involved in politics (Kanpō 1940a, p. 40)". In response, on the following day, Army Minister Hata Shunroku 畑俊六 claimed that "the New Order in East Asia will be established to manifest the great cause of *Hakkō Ichiu*, which has been the national enterprise since the founding of Japan. This is why it is called a holy war and is fundamentally different from the so-called aggression war, which is just the law of the jungle (Kanpō 1940b, p. 46)". Saitō was one of the rare wartime anti-militarist politicians and one of the staunchest defenders of particracy. He was soon expelled from the Diet for his blasphemy, which also goes to show that the expansionist reading of Jimmu's ascension to the Throne had become inviolable as a principle governing national policy (Edwards 2003, p. 311).

### 4.2. *Hakkō Ichiu and the Military Spirit*

In fact, the term *Hakkō Ichiu* had long been popular in both civilian and military communities. The Manchurian Incident of 1931 was a catalyst for the spread of *Hakkō Ichiu* among the military, a fact that was also foreshadowed by the Showa Restoration, with a series of coup d'états based on the spirit of Jimmu's national foundation. In particular, the *Kekki Shuisho* 「蹶起趣意書」 ("Manifesto of the Uprising"), which the young military officers wished to submit to the Emperor in the 26 February Incident in 1936, advocated: "Under the leadership of His Majesty the Emperor, the whole nation shall live and grow in unity, and the national spirit of *Hakkō Ichiu* shall be fully realized (Kuroiwa 2022, p. 11)" Throughout the 1930s, young military officers who aspired to the Showa Restoration and participated in several assassination attempts were more or less influenced or were believers in Nichirenism. Nevertheless, in this context, the use of *Hakkō Ichiu* was not so much about expansionism than about the return to Jimmu's spirit.

However, from the late 1930s onwards, *Hakkō Ichiu* became increasingly used in public life to glorify wars of aggression. After the government decided to launch the National Spiritual Mobilization Movement, the Cabinet Intelligence Bureau immediately sponsored a competition to solicit, from all over the country, that a march be composed, so as to express the ideals of the empire and the spirit of the nation. It was scheduled to be broadcasted on the "National Ballad," a popular national radio program. The song, entitled *Aikoku Kōshinkyoku* 愛国行進曲 ("Patriotic March"), was released in December 1937. It soon became popular music, selling more than 100,000 copies within a few days, the figure rising to a million shortly thereafter (Oba 2002, p. 234). In the lyrics, it portrayed the glory of the emperor and the mission of the subjects in quaint verse:

> "He who reigns above in power and in virtue dight.
> Sovereign of unbroken line is our changeless light.
> We will follow—one and all loyal subjects, we—
> Follow Him aright: fulfil our great destiny!
> Onward, east, west, north, and south. Over land and main!
> Let us make the world our home, call to fellow men (*Yuke Hakkō wo Ie to nashi* 往け八紘を宇となし).
> Everywhere on the four seas, let us build the tower of just peace—let our ideal
> Bloom forth like a flower." [13]

Above are the lyrics of the song's second verse, in which the emperor is like the shepherd leading his flock to peace and light. The sixth line (the fifth line in the Japanese version) is literally translated as "Carry up the eight cords (*Hakkō*) to be the roof", and it is

a variant of *Hakkō Ichiu*, expressing the ideal of universal brotherhood under the emperor. *Hakkō Ichiu* seemed to be a rallying cry for more peoples to join the emperor's flock. As argued by Atsuko Ichijo, the Meiji oligarchs believed that religion was essential for the building of a modern nation: "The Meiji oligarchs thought that something similar to Christianity was essential to mold the people of Japan into a unified and self-aware Japanese nation (Ichijo 2009, p. 126)". Thus, in a way, State Shinto was a product of a certain interpretation of Christianity, elevating the emperor and the imperial family to an exalted position close to God.

During the war, mass media were increasingly used for war propaganda, and newsreels were becoming a novel form of broadcast that allowed audiences to watch carefully censored videos of the war front. Every newsreel released in Japan from 1940 to 1945 began and ended with a brief flickered trademark of *Hakkō Ichiu* (Figure 1), such as MGM's lion or Paramount's mountain and stars. The logo consisted of a golden kite spreading its wings in an attempt to encircle the Earth that emitted rays from behind, and the image of the Earth precisely showed the entirety of East Asia. Roger W. Purdy argued that "it projected to theater audiences throughout Tokyo's empire a gleaming vision of a new East Asia enlightened and protected by Japanese Leadership (Purdy 2009, pp. 106–7)." Moreover, it had a strong connection with the mythology of Emperor Jimmu: according to the *Nihongi*, when Emperor Jimmu struggled with Nagasunehiko, a golden kite landed on the tip of Jimmu's bow, the light emitted from the kite's body blinded Nagasunehiko's soldiers and enabled the army of the eastward expedition to win the war (Translation by Aston 1972, p. 126). The golden kite thereby became the symbol of Jimmu's victory and appeared everywhere, such as on medals, stamps, and cigarette cases in wartime Japan. If *Hakkō Ichiu* was the linguistic rhetoric of the State Shinto myth, then the golden kite was its pictorial rhetoric. Consequently, with its figurative forms, the trademark of the newsreel was a perfect visualization of the origins and ambitions of the Imperial Japanese *Hakkō Ichiu*.

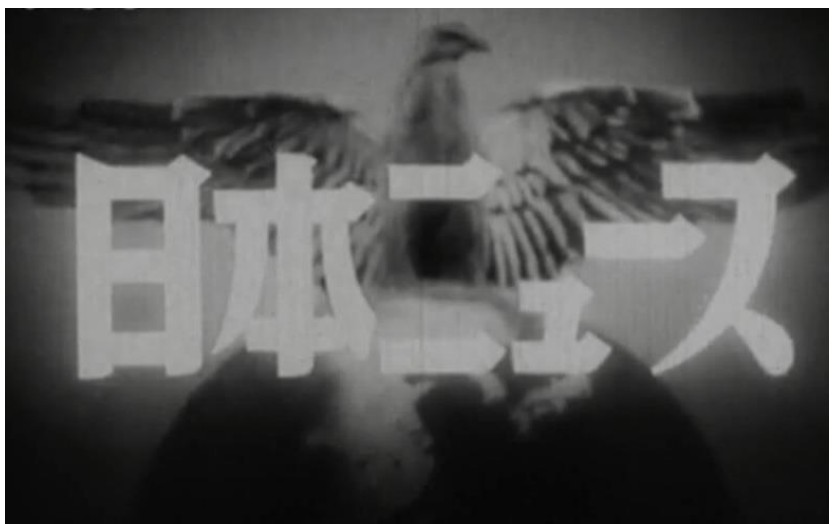

**Figure 1.** Japanese newsreels' logo. © NHK Archives. 2022.

## 5. The 2600th Anniversary of the Empire and the Hakkō Ichiu Tower

When living in the Japanese Empire, one would orient oneself in time according to two types of chronology in parallel: firstly, the regnal year chronology based on the reigning emperor, such as Meiji, Taishō, and Shōwa; secondly, the *Kōki* 皇紀 (Japanese imperial year), counting years from the enthronement of Emperor Jimmu in 660BC, the legendary time of the state's foundation.[14]

### 5.1. Planning for the Anniversary

Accordingly, in 1940, the Japanese celebrated the 2600th anniversary of the Empire via a series of celebrations, a climactic moment for the "unbroken imperial line" and *Kokutai* ideology, essential to emperor worship and imperial legitimacy until defeat in 1945 (Ruoff 2010, p. 1). The Japanese made a bid to host the 1940 Summer Olympics, in conjunction with the 2600th anniversary celebrations, as early as the early 1930s. During the Berlin 1936 Summer Olympics, Tokyo was successfully chosen as the 1940 Olympic Games host city by the International Olympic Committee (IOC). The same year, Japan planned to hold the Sapporo Winter Olympics and the Tokyo World EXPO to commemorate its twenty-sixth centennial. However, with Japan stuck in the Sino-Japanese War, these world events were canceled, and the hosting rights were handed back to the IOC. Thus, the commemoration of the 2600th anniversary could only be limited to the domestic scope.

In October 1935, the government established the 2600th Anniversary Events Bureau to plan and promote commemorative events, including the renovation of the Kashihara Shrine and Jimmu's mausoleum, and, in 1937, they established the semigovernmental, semicivil "Association to celebrate the 2600th Anniversary", which would organize fundraising events (Ruoff 2010, p. 13). In 1940, various commemorative events were held throughout the country and colonies, including exhibitions and athletic meetings, and, on 11 February, *Kigensetsu* (the festival for celebrating Jimmu's enthronement), more than 110,000 shrines nationwide practiced rituals. The commemoration culminating on 10 November, the 2600th Anniversary Ceremony was grandly held by the second Konoe cabinet in front of the Tokyo Imperial Palace, over which Emperor Hirohito and Empress Kōjun presided, and related events continued to be held until 14 November. Kenneth J. Ruoff believes that, among modern nations, only the 2500th anniversary of the founding of the Persian Empire staged by Iran in 1971 could be comparable to Japan's commemoration in terms of the outrageous extent of historical continuity attributed to what was a modern nation state and also in the crediting of the monarchy for this lengthy continuity and unity (Ruoff 2010, p. 1). In addition to the national level, a series of activities had been carried out at the local level to commemorate the 2600th anniversary, the most famous of which was the construction of the Hakkō Ichiu Tower (Figure 2) in Miyazaki Prefecture.

### 5.2. A Tower to Be Erected

As mentioned above, Miyazaki Prefecture, formerly known as Hyūga, was believed to be the birthplace of Emperor Jimmu and the origin of the sacred imperial state of Japan in State Shinto mythology. Thus, local residents called their prefecture their "ancestral land", considered Miyazaki Prefecture to be a gift for the whole of Japan, and argued that, somehow, Miyazaki was an ancestral land for the rest of the country as well. Some scholars called this regional nationalism "*Sokoku Hyūga Shugi*" 祖国日向主義 (Ethnocentrism of Hyūga) (Kuroiwa 2022, p. 163). In 1937, Aikawa Katsuroku 相川勝六, a self-proclaimed kami-fearing man and fervent nationalist, was appointed as the new governor of Miyazaki Prefecture. In response to the National Spiritual Mobilization Movement and the regional nationalistic sentiment of Miyazaki, Aikawa organized a number of activities, including the establishment of the "Ancestral Land Hyūga Promotion Unit (Hakkō 2017, pp. 74–76)". However, his greatest ambition was to erect a monument in Miyazaki that would become a landmark for the country to commemorate the spirit of Emperor Jimmu's national foundation. On 2 December 1938, he unveiled the plan to build the Hakkō Ichiu Tower in such a way as to coincide with the commemoration of the 2600th anniversary at the Prefectural Assembly. He argued that, as the prefecture's own project, it should be the highest tower in Japan, adorned with the motto of *Hakkō Ichiu*. He then laid out the basics of his plan: "We will select a suitable site in the site of Emperor Jimmu's palace prior to eastern expedition, and erect the magnificent Pillar of Heaven in the purely Japanese style, made of solid stones (Hakkō 2017, p. 82)." On the following *Kigensetsu*, 2 February 1939, the *Ōsaka mainichi* published an article requesting assistance in terms of donations of money and stones from all communities. On the same day, Miyazaki Prefecture also

released the outline of the 2600th anniversary celebration, emphasizing that the stones donated for the construction came not only from East Asia but also from every corner of the world where the Japanese had reached. This would contribute to highlight the achievements of the present emperor's holy war and to promote the spirit of *Hakkō Ichiu* originated by Emperor Jimmu (Hakkō 2017, p. 88).

Consequently, it solicited 1789 standard-size stones,[15] used in the tower's base, with the vast majority of stones coming from the local area of Miyazaki and domestic Japan, donated from various governmental and civic institutions, such as Miyazaki Prefecture Women's Normal School, Fukui Prefecture Shrine Association, and so forth.[16] Aikawa also asked War Minister Itagaki Seishirō 板垣征四郎 and Sugiyama Hajime 杉山元, the commander of the North China Area Army, to cooperate in soliciting stones, so the imperial troops plundered booty from all over China from the so-called Manchukuo–Soviet border in the north to the Mongolian border in the west to Hainan Island in the south, and brought them back to Miyazaki, even looting buildings of historical value, including the stone from Ming Palace in Nanjing, just one year after the massacre (Hakkō 2017, pp. 54–55, 92). There were also a few stones donated by Japanese groups from Southeast Asia, the United States, Germany, South America, etc. The spoliatory stones from China and colonies were indeed a good demonstration of the spirit of *Hakkō Ichiu*. Still, they also became incriminating evidence of Japan's aggression and expansion.

The famous sculptor Hinago Jitsuzo 日名子実三, who sculpted a statue of Nichiren and had been influenced by Tanaka, volunteered to design and carve this tower without receiving payment (Kuroiwa 2022, pp. 36–38). When he visited the Miyazaki Shrine, he saw the *Gohei* 御幣[17], and gained inspiration from it, combing it with a shield in his design. During the construction of the tower, several local civic groups, such as the Patriotic Women's Association and Ancestral Land Hyūga Promotion Unit, were mobilized to participate in the construction.

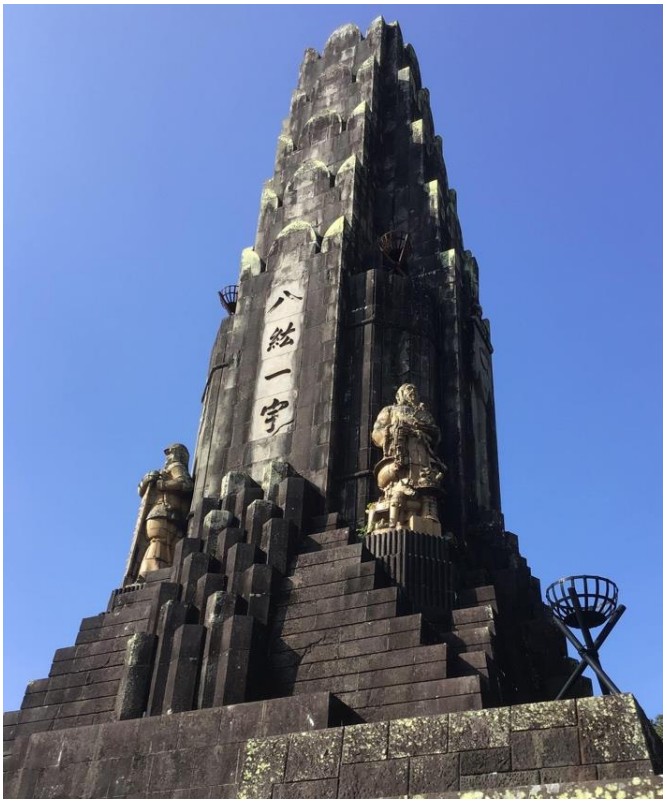

**Figure 2.** Hakkō Ichiu Tower, Miyazaki, Japan. © Hatena Blog. 2022.

### 5.3. A Monumental Rhetoric

The tower was 36.4 meters high, with a square base of 65 meters in circumference, made of stacked stones at the bottom. On the frontage of the tower was the four-character *Hakkō Ichiu* inscription written by Prince Yasuhito Chichibu. This is why the tower is commonly known as *Hakkō Ichiu no Tō* 八紘一宇の塔, although its official name was *Ametsuchi no Motohashira* 八紘之基柱. On the back of the tower was carved the overview of the Japanese Empire, manifesting the power of the empire (Hakkō 2017, pp. 16–24). At the four corners of the base of the tower were braziers, which represent the unity and endeavor of the people, and beneath the braziers were four *kami* (Aratama, Nigitama, Sachitama, and Kushitama) symbolizing courage, peace, love, and wisdom in their mythological images, which were also representations of warriors, craftsmen, farmers, and fishermen. Inside the tower was an adytum (the architect Hinago believed that the soul of the tower lies in the interior). On the bronze door of the adytum was engraved an etching of Jimmu's eastern expedition, with the army sailing from Hyūga, warships in formation, warriors holding flags and rowing oars, an image of irresistible momentum. Inside the adytum was the calligraphy of the expression *Hakkō Ichiu* by Prince Yasuhito Chichibu.

The tower was finally completed on 25 November 1940, in time for the 2600th anniversary of Jimmu's ascension. Prince Chichibu was not present due to health reasons, and the third younger brother of Emperor Shōwa, Prince Nobuhito Takamatsu, presided over the ceremony instead. The ceremony can be seen as the *fin de siècle* entertainment of Imperial Japan, as the catastrophe came a few years later.

During wartime, the image of the Hakkō Ichiu Tower as a symbol of ultranationalism was popular in the Empire, appearing on stamps, postcards, and even printed on the 10-yen bill. As a monument of State Shinto, the tower was undoubtedly the most substantive expression of *Hakkō ichiu* through its architectural rhetoric, fusing the mythology of Shinto, the ideology of *Kokutai*, and the imperial warfare phraseology. Meanwhile, it also manifested a regional nationalism promoted by Governor Aikawa. The stones donated from all over the world, especially from the occupied areas and colonies, gave its material expression to the expansionism represented by the *Hakkō Ichiu* discourse.

### 6. Conclusions

The *Hakkō Ichiu* rhetoric is imbued with a diversity of sources and expressions, interacting among themselves. It refers to a cosmic spatial apprehension typical of ancient East Asian thought. It is also closely related to the Shinto mythology of Emperor Jimmu's eastward expedition. In the ideology of State Shinto, the emphasis on the myth of Emperor Jimmu also evolved from the original *Tenjō Mukyū*, which represented the unbroken Imperial line, to *Hakkō Ichiu*, representing Japan's imperial expansionism, enriching but also displacing the connotations attached to Jimmu's legend in State Shinto during the early 20th century. It, thus, provided state legitimacy to a policy of foreign aggression.

Additionally, one should not ignore the contribution of Tanaka Chigaku, who rediscovered *Hakkō Ichiu* in the modern era and gave it renewed expression and meaning. Tanaka, a firm nationalist, creatively expounded Nichiren's thought, launching a Buddhist Renaissance movement that he called Nichirenism. After realizing that *Kokutai* offered a solid foundation for Japanese nationalism, he intelligently exploited it, and the mythology of Shinto became an excellent ideological resource for the promotion of Nichirenism. It is at this point that he rediscovered the *Hakkō Ichiu* phraseology and gave it an expansionist coloring, through which he depicted the Buddhist utopia of world unification. Tanaka's contribution allowed for the reviving of a syncretistic approach to the worship of both *kami* and buddhas, this after the syncretism underwent a brief hiatus during the Meiji Restoration period. Although State Shinto was not a religion in the historical context, Tanaka still fostered Nichirenism to be one of the most important nationalist forces in Imperial Japan, by incorporating elements from State Shinto and *Kokutai*. Furthermore, Nichirenism profoundly influenced the nationalist movement in the early Showa era, and many military

officers, including Ishiwara Kanji, were followers of Nichirenism, whose radical actions accelerated Imperial Japan into disastrous wars.

After the Second Sino-Japanese War, the *Hakkō Ichiu* phraseology escaped the context of religious discourse and became a tool utilized in order to glorify the war of aggression and colonization. Its use illustrated the ideology of emperor worship and continuous imperial expansion, becoming synonymous with militarism and ultranationalism. Because of its widespread use in war propaganda, it was well known to the public through a wide range of embodied and material representations, including lyrics, newsreels, stamps, bills, and even architecture. The Hakkō Ichiu Tower, built to commemorate the 2600th anniversary of national foundation, played a special role in perpetuating this State Shinto rhetoric. In addition to the Hakkō Ichiu Tower, the motto was often carved on stone monuments and pillars of shrines throughout Japan and overseas, manifesting the ideal of "universal brotherhood".

The term *Hakkō Ichiu* was banned by the GHQ through the "Shinto Directive"[18] as a term promoting State Shinto, militarism, and ultranationalism. In 1946, the GHQ ordered the removal of the inscription *Hakkō Ichiu* on the front face of the tower as well as the one of the statue of Aratama, the symbol of the warriors. Subsequently, the tower was left unattended. However, for the sake of tourism promotion, the Prefecture gradually proceeded to readminister the tower and its surrounding area. The place was renamed "Peace Tower" in 1957, the statue of Aratama was restored in 1965, and the motto *Hakkō Ichiu* was resharpened in the same year. Despite opposition from left-wing groups, supporters claimed that: "*Hakkō Ichiu* conveys the idea of praying for world peace and has nothing to do with the war (Kuroiwa 2022, pp. 143–44)". Today, the nationalistic rhetoric of *Hakkō Ichiu* could be compared to a nearly dead language, both in the Shinto context and in public life. However, its historical and religious connotations still deserve scrutiny.

**Funding:** This research received no external funding.

**Institutional Review Board Statement:** Not applicable.

**Informed Consent Statement:** Not applicable.

**Data Availability Statement:** Not applicable.

**Conflicts of Interest:** The author declares no conflict of interest.

## Notes

1.  Some scholars literally translated *Hakkō Ichiu* as "eight corners of the world under one roof" (Beasley 1987, p. 226; Edwards 2003, p. 291), but R. W. Purdy translated it as "eight cords, one roof", and I prefer Purdy's translation, "eight cords" (Purdy 2009, p. 106), because it better reflects the original meaning of the word from an etymological perspective. In the second section, I discuss how "eight cords" can refer to the world.

2.  Inoue Hiroshi's summary is quoted in Okuyama Michiaki's paper (Okuyama 2011, p. 135).

3.  八者，維綱也。天地以發明，故聖人以合陰陽之數也, see (Fang 2008, p. 1287).

4.  "Four corners are called *Wei*; four directions are called *Gang*." (四隅曰维，四正曰纲.) (Fang 2008, p. 1292)

5.  紘，維也。維落天地而為之表，故曰紘也 (He 1998, p. 334).

6.  It is noteworthy that the applicability of Shinto to the concept of "religion" has been controversial since it was introduced to Japan along with a number of Euro-American concepts in the 1870s (Josephson 2012, p. 94). The bureaucrats in the Meiji era strove to draw a clear boundary between the secular and religious spheres, avoiding to define State Shinto as a religion and especially a state religion, but rather the rites of state, a patriotic morality in which all people were compelled to participate in. The distinction between State Shinto and other religions, such as Buddhism, Christianity and Sect Shinto, was eventually confirmed in the 1889 Imperial Constitution, as Trent E. Maxey commented, "the Constitution codified the religious settlement by explicitly rejecting religion as a component of national definition. It thus adopted the principle of religious freedom over toleration (Maxey 2014, p. 14)." This constituted what Yasumaru Yoshio calls the "Separation of church and state of Japanese type", in which State Shinto was the rites of state in the public sphere requiring mandatory participation, while the religious affairs were restricted to the private sphere, and individuals had the constitutional right of the freedom of belief (Yasumaru 1979, pp. 208–9). However, it cannot be ignored that the distinction was more confined to the legal and administrative level. State Shinto contained many religious elements, from historical sources and mythology to the ritual with its temporal and spatial dimensions. In reality, there

were still multiple cases of conflicts between State Shinto and religious beliefs, especially in the area of individual spirituality, such as the lèse-majesté incident of Uchimura Kanzo and the 1932 Sophia University—Yasukuni Shrine incident. Therefore, this article prefers to define State Shinto as a "quasi-religion" and discusses its rhetoric with a religious dimension.

7. It was often articulated as Jimmu's entrepreneurship spirit (神武創業の精神) or spirit of national foundation (肇国の精神). During the Meiji Restoration period, the promotion of Jimmu's spirit was often associated with the abolition of the shogunate system and direct imperial rule, but its exact meaning was still left vague.

8. The phrase was derived from the imperial edict of *Tenjō Mukyū* (天壤無窮の勅令) (Toneri 2019, pp. 32–33).

9. If used as a term of Religious Studies, Ōtani Eiichi defined Nichirenism as "the Nichirenism was a social and political movement in pre-World War II Japan, aiming to achieve a utopian world through the unification of Japan and the unification of the world through the Buddhist unity of government and religion based on the *Lotus Sūtra*. It is a Buddhist religious movement developed with ambition." Qtd. (Ōtani 2001, p. 15.)

10. Yamato is another name for Japan, which includes the pronunciation of the word mountain, "Yama."

11. For more information on Ishiwara's ideas of Nichirenism and the final war, as well as the practice, see (Godart 2015).

12. At that time, neither side of the conflict declared war, and the Nationalist government of China did not formally declare war until after the Pearl Harbor attack in 1941, while Japan called it "collision" or *Hokushi Jihen* 北支事变 (North China Incident) domestically, and only after the Battle of Shanghai did the word *Sen* 戦 (war) appear, which are all called "war" in this paper.

13. The official English lyric was translated by Foreign Ministry officer Obata Shigeyoshi 小畑薫良, see (Obata 1938, p. 27).

14. Although *Anno Domini* was introduced to Japan as early as the Meiji era, it was not as commonly used as in other Asian countries prior to 1945. Additionally, it is often translated as *Seireiki* 西暦 (Western Year).

15. According to the statistics from the Hakkō Ichiu Tower Research Association, they collected 1789 pieces, totaling 834 stere of stone (Hakkō 2017, p. 18).

16. Walter Edwards gives a table of donors for stones (Edwards 2003, pp. 297–98).

17. A wooden wand with two zigzagging paper streamers used in Shinto rituals to bless or sanctify people or objects.

18. The Shinto Directive was an order issued by the GHQ to abolish State Shinto and Japanese ultranationalistic and militaristic slogans in 1945, and its full title was "Abolition of Governmental Sponsorship, Support, Perpetuation, Control and Dissemination of State Shinto" (SCAPIN-448 1945, p. 3).

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
