# Peer review of "Hakkō Ichiu: Religious Rhetoric in Imperial Japan"

_religions, doi:10.3390/rel14010021_

Round 1
Reviewer 1 Report
An excellently written and highly informative article, which clearly articulates the interrelationship between religion and politics in Meiji Japan and provides detailed background information regarding the development of the key religious concepts that were mobilized during this era’s construct of Japanese nationalism.
Minor points:
In many places, your hyphens will need to be replaced with en-dashes or even em-dashes. Check with the editor.
Meiji Restoration >> add dates
complied >> compiled
Japanese held grandiose >> The Japanese held grandiose
based on searching for evidence of Japanese war of aggression and colonialism, holding several exhibitions in China >> the sentence could be improved for clarity, perhaps by shortening it somewhat. Note also: add the, i.e. the Japanese war of
more or less synonym to >> more or less synonymous to
What are the difference >> plural
summarized in introduction >> summarized in the introduction
was eager to seek the new regime’s legitimacy >> was eager to establish the new regime’s legitimacy
in work by the Meiji era historian >> in a work by the Meiji era historian
How felicity >> How fortunate that
but also a Canon >> but was also a canon
by the Nichiren Buddhism preacher >> by the Nichiren Buddhist preacher
he was ten-year-old >> he was ten years old
lay Buddhism campaigns >> lay Buddhist campaigns
Tanaka believed >> Tanaka believed, in line with Nichiren Buddhist teachings, that
cosmopolitanism view >> cosmopolitan view
Chakravartin, dharmachakra >> correct spelling is cakra
passed down till now >> passed down to the present
It was representing >> It represented
he was also endeavouring >> he also endeavoured
albeit Tanaka himself >> although Tanaka himself
did not mean the same thing as having a foreign hegemonic state annexing the territory of another country >> How this relates to the Japanese government’s expansionism may be to be further clarified.
For tackling the second problem >> To tackle the second problem
to the one on >> delete ‘the’
would later on changed for >> later on be changed for
which also showed the >> which also goes to show that the
the entire East Asia >> the entirety of East Asia
German, South America >> Germany
albeit its official name >> although its official name
The Hakkō Ichiu Rhetoric >> The Hakkō Ichiu rhetoric
Lotus Sutra >> Lotus Sūtra [sutra is okay, but sūtra is the correct spelling]
other Sutras and other sects >> other sūtras and other sects
Buddhism and Shinto sharing religious features >> Buddhism and Shinto to share religious features
of the warriors, the tower began subsequently left unattended. >> of the warriors. Subsequently, the tower was left unattended.
the motto Hakkō Ichiu was resharpened >> Hakkō Ichiu italics. Also, I’m not sure that ‘resharpened’ is the right word. Reinterpreted?
in public life; However >> full stop
Author Response
Reply to Comments on
Manuscript religions-2070449:
“Hakkō Ichiu: Religious Rhetoric in Imperial Japan”
Dear Editors and Reviewers:
The revised version of the manuscript (Manuscript religions‑2070449) has been submitted, which has been cautiously revised according to the reviewers’ comments.
I would like to express my sincere gratitude to Editors and three anonymous reviewers for their precious time and constructive suggestions. Their insightful comments are very helpful in improving the quality and presentation of this paper. All the comments have been seriously considered and addressed in the revised manuscript.
Below is an item-by-item response to the comments from three anonymous reviewers.
Reviewer 1’s Comments
An excellently written and highly informative article, which clearly articulates the interrelationship between religion and politics in Meiji Japan and provides detailed background information regarding the development of the key religious concepts that were mobilized during this era’s construct of Japanese nationalism.
Minor points:
In many places, your hyphens will need to be replaced with en-dashes or even em-dashes. Check with the editor.
Meiji Restoration >> add dates
complied >> compiled
Japanese held grandiose >> The Japanese held grandiose
based on searching for evidence of Japanese war of aggression and colonialism, holding several exhibitions in China >> the sentence could be improved for clarity, perhaps by shortening it somewhat. Note also: add the, i.e. the Japanese war of
more or less synonym to >> more or less synonymous to
What are the difference >> plural
summarized in introduction >> summarized in the introduction
was eager to seek the new regime’s legitimacy >> was eager to establish the new regime’s legitimacy
in work by the Meiji era historian >> in a work by the Meiji era historian
How felicity >> How fortunate that
but also a Canon >> but was also a canon
by the Nichiren Buddhism preacher >> by the Nichiren Buddhist preacher
he was ten-year-old >> he was ten years old
lay Buddhism campaigns >> lay Buddhist campaigns
Tanaka believed >> Tanaka believed, in line with Nichiren Buddhist teachings, that
cosmopolitanism view >> cosmopolitan view
Chakravartin, dharmachakra >> correct spelling is cakra
passed down till now >> passed down to the present
It was representing >> It represented
he was also endeavouring >> he also endeavoured
albeit Tanaka himself >> although Tanaka himself
did not mean the same thing as having a foreign hegemonic state annexing the territory of another country >> How this relates to the Japanese government’s expansionism may be to be further clarified.
For tackling the second problem >> To tackle the second problem
to the one on >> delete ‘the’
would later on changed for >> later on be changed for
which also showed the >> which also goes to show that the
the entire East Asia >> the entirety of East Asia
German, South America >> Germany
albeit its official name >> although its official name
The Hakkō Ichiu Rhetoric >> The Hakkō Ichiu rhetoric
Lotus Sutra >> Lotus Sūtra [sutra is okay, but sūtra is the correct spelling]
other Sutras and other sects >> other sūtras and other sects
Buddhism and Shinto sharing religious features >> Buddhism and Shinto to share religious features
of the warriors, the tower began subsequently left unattended. >> of the warriors. Subsequently, the tower was left unattended.
the motto Hakkō Ichiu was resharpened >> Hakkō Ichiu italics. Also, I’m not sure that ‘resharpened’ is the right word. Reinterpreted?
in public life; However >> full stop
Respond:
I would like to thank you very much for scrutinizing my paper very carefully and giving a number of comments about the syntax and typos. In particular, I benefited a lot from several Buddhist terms. I accepted basically all the points and modified them in my paper, which made my paper more fluent and comprehensible.
The only reservation I have is about the word “resharpened”. The motto Hakkō Ichiu was first worn away on the monument by GHQ, and then re-carved in 1965, so in my opinion the word “resharpened” makes sense.
Thank you again for your precious time and valuable comments.
Author
2022/12/12

Reviewer 2 Report
This is a really interesting and well-researched article on an important and fascinating topic. On the whole the article works as it stands. It has a solid structure and the argument is well-founded. I would recommend one more language check though, there are still some minor mistakes to grammar and choice of vocabulary that could be amended. These are minor though, the language on the whole is very good.
There is one major issue though that requires some revision. The author relies on some very outdated sources when trying to make sense of “State Shinto”, and this works to the detriment of the whole text. Some significant points here are that a) State Shinto was never a religion and most importantly never a state religion, b) the 1890 constitution created a framework for understanding religion that was written as a direct response to social debates about the place of the emperor, and c) mandatory shrine rites and emperor worship were a part of life for everyone in the empire, and not something that one could choose to adhere to. This was all a conscious part of the system. There was religious freedom, but there was also expressing national identity.
My main suggestion is that the author reads Trent E. Maxey’s “The Greatest Problem” (2014). The question of how the categories of Shinto and religion worked under Japanese law is important, and the fact that the author mostly ignores this topic distracts from an otherwise really well put-together text. I have some other alternative reading suggestions in the notes below, but I think Maxey would be the most relevant work for what this article attempts to do.
p. 1: ”State Shinto” needs to be nuanced. In what way was State Shinto “established as the official ideology” in Imperial Japan? If you use Murakami Shigeyoshi’s or Shimazono Susumu’s definition of the term, make this clear. This is contested territory. I return to this below.
p. 3: “The word Hakkō first appeared…” Confusing use of the Japanese reading, should be Bahong here?
pp. 4-5: The narrative of State Shinto here is based on seriously outdated research and would do well with an update. I would suggest looking at Trent E. Maxey’s “The Greatest Problem” (2014) for a discussion of the relationship between how religion and Shinto was created and the drafting of the 1890 constitution. But really, just relying on the Hardacre’s most recent book (“Shinto: A History”, 2017), already cited in the article, would work better here. There is a lot more nuance to this story than the what is presented in this text, and it is important to the argument made in the article. Another suggestion might be the work of Isomae Jun’ichi, almost anything would work but one suggestion is Shūkyō gainen arui wa shūkyō-gaku no shi 宗教概念あるいは宗教学の死 (2012). Isomae has used Hobsbawm’s notion of invented tradition to explore the formation of State Shinto and all its rites and festivals.
p. 5: “the GHQ” – give us the full name the first time it is used, and maybe contextualize it. Not everyone is familiar with how the occupation was organized.
p. 5: “Nation-Body” – Kokutai is very commonly translated as “national polity” in English. Not saying the author has to follow this pattern, but it is well established.
p. 7: The narrative presented here about Tanaka’s wish to make Nichirenism a “state religion” in Japan would really benefit from a better contextualization, in particular with regards to “religion” under the 1890 constitution. I really think the author should look at Maxey’s work (see above).
p. 8: “As scholar Yulia Burenina commented” – fairly superfluous use of “scholar”, the reader assumes that those cited are scholars, unless otherwise noted.
p. 11: “far from the animistic Shinto with belief in kami (spirits)” – referring to Shinto as “animist” is stepping into an unnecessary and deeply polemic discussion about the use of “animism” in the study of religion in Asia. There is no need to do so here, so I would just avoid the term. Also, the discussion in this paragraph would again benefit from a better understanding of the constitutional debates that resulted in Shinto being placed outside of the discourse of “religion” in Meiji Japan. Maxey’s work is really good here, but perhaps the author should also consider Jason Josephson Storm’s “The Invention of Religion in Japan” (2012), as a way of deepening the discussion.
p. 11: “Scholar Purdy argued…” – Again, unnecessary use of “scholar” here.
p. 12: “When living in the Japanese Empire, one would orient oneself in time according to three types of chronology in parallel: firstly, the Common Era chronology…” – this is not correct, right? While the common era was used by some (in particular Christians) prior to 1945, the reign era system would have been fully dominant in Imperial Japan.
p. 13: The project of building the tower using resources from the whole empire as described here reminds me of how the Great Buddhas in Nara and Kamakura were funded and built. Was there a conscious parallel to these more ancient national projects as well?
p. 15: “Tanaka was not a follower of State Shinto but a firm nationalist” – this sentence is incomprehensible. State Shinto, if that is what the author wants to call the “secular” system of mandatory shrine rites and emperor worship prior to 1945, was not something you could “choose” to follow. All imperial subjects were part of this system, just as they were part of the Empire of Japan. Religion had followers, “State Shinto” did not. It only had imperial subjects. Again, I think Maxey’s work would be useful here.
p. 15 (cont.): The discussion about “syncretism” is also a bit off. Buddhism and mandatory shrine rites coexisted throughout most of the Meiji period, and through all of Taisho and imperial Showa. Under the 1890 constitution Buddhism was simply compartmentalized in its own sphere as a “religion” and was therefore optional, whereas shrines were not. Tanaka didn’t question the emperor’s divinity, because to do so would be lèse-majesté. He wanted a place for Buddhism in the mandatory national narrative, he did not “reinvent” syncretism. Again, Maxey discusses exactly these debates.
Author Response
Reply to Comments on
Manuscript religions-2070449:
“Hakkō Ichiu: Religious Rhetoric in Imperial Japan”
Dear Editors and Reviewers:
The revised version of the manuscript (Manuscript religions‑2070449) has been submitted, which has been cautiously revised according to the reviewers’ comments.
I would like to express my sincere gratitude to Editors and three anonymous reviewers for their precious time and constructive suggestions. Their insightful comments are very helpful in improving the quality and presentation of this paper. All the comments have been seriously considered and addressed in the revised manuscript.
Below is an item-by-item response to the comments from three anonymous reviewers.
Reviewer 2’s Comments
This is a really interesting and well-researched article on an important and fascinating topic. On the whole the article works as it stands. It has a solid structure and the argument is well-founded. I would recommend one more language check though, there are still some minor mistakes to grammar and choice of vocabulary that could be amended. These are minor though, the language on the whole is very good.
There is one major issue though that requires some revision. The author relies on some very outdated sources when trying to make sense of “State Shinto”, and this works to the detriment of the whole text. Some significant points here are that a) State Shinto was never a religion and most importantly never a state religion, b) the 1890 constitution created a framework for understanding religion that was written as a direct response to social debates about the place of the emperor, and c) mandatory shrine rites and emperor worship were a part of life for everyone in the empire, and not something that one could choose to adhere to. This was all a conscious part of the system. There was religious freedom, but there was also expressing national identity.
My main suggestion is that the author reads Trent E. Maxey’s “The Greatest Problem” (2014). The question of how the categories of Shinto and religion worked under Japanese law is important, and the fact that the author mostly ignores this topic distracts from an otherwise really well put-together text. I have some other alternative reading suggestions in the notes below, but I think Maxey would be the most relevant work for what this article attempts to do.
Respond: I really appreciate your comments, and the Trent E. Maxey’s book “The Greatest Problem”: Religion and State Formation in Meiji Japan that you recommended, which are very constructive in enhancing my article, and helping me with my subsequent studies. State Shinto is a very contentious topic, and I understand that it was not a religion, especially a state religion in Imperial Japan. The government sought to draw the boundary between religious and secular, and regarded State Shinto as “rites of state” in Meiji era. However, I think that State Shinto contained some elements of a religious dimension, so it’s possible to analyze its rhetorical approach from this perspective.
Again, I am sincerely grateful for your precious time and comments, and following are the responses to the comments.
p.1: “State Shinto” needs to be nuanced. In what way was State Shinto “established as the official ideology” in Imperial Japan? If you use Murakami Shigeyoshi’s or Shimazono Susumu’s definition of the term, make this clear. This is contested territory. I return to this below.
Respond: According to your comment, I have added a more detailed literature review of the concept of State Shinto:
“From Murakami Shigeyoshi’s influential book, Kokka Shintō onwards, scholars have developed successive discourses around the idea of State Shinto and its influence on postwar Japan. Murakami regarded State Shinto as a state religion, which led the country into militarism, ultranationalism and disastrous wars, by ruling the spiritual world of Japanese people (Murakami 1970). Shimazono Susumu emphasizes State Shinto’s in-fluence on education, and regards the imperial rites as an important part of State Shinto, which has not been fundamentally abolished to date (Shimazono 2010, 2021). Inoue Hiroshi regards State Shinto was modern Japan’s system of official religion in a form that emerged from the theoretical and institutional reorganization of shrines and “Shinto”. Some western researchers prefer to treat State Shinto as an “invented tradition”. Helen Hardacre analyzes how Shinto formed new relations with government through state management, which deeply affected shrines, the priesthood, and shrine communities (Hardacre 1989, 2017). Trent E. Maxey expounded how the Meiji government strove to draw a clear line of demarcation between secular and religious dominations (Maxey 2014). In recent years scholarship has gradually focused on the relationship between Kokutai 国体 (“national polity”) and State Shinto, a dimension that had been overlooked in past historical studies. In particular, in 2019, Fudita Hiromasa compiled the results of a symposium, State Shinto and Kokutai: A study of the interdisciplinarity of Religion and Na-tionalism, advancing research in the field (Fudita 2019).”
p.3: “The word Hakkōfirst appeared…” Confusing use of the Japanese reading, should be Bahonghere?
Respond: I accepted your comment, and change the expression Hakkō to Bahong in this sentence.
p.4-5: The narrative of State Shinto here is based on seriously outdated research and would do well with an update. I would suggest looking at Trent E. Maxey’s “The Greatest Problem” (2014) for a discussion of the relationship between how religion and Shinto was created and the drafting of the 1890 constitution. But really, just relying on the Hardacre’s most recent book (“Shinto: A History”, 2017), already cited in the article, would work better here. There is a lot more nuance to this story than the what is presented in this text, and it is important to the argument made in the article. Another suggestion might be the work of Isomae Jun’ichi, almost anything would work but one suggestion is Shūkyō gainen arui wa shūkyō-gaku no shi宗教概念あるいは宗教学の死(2012). Isomae has used Hobsbawm’s notion of invented tradition to explore the formation of State Shinto and all its rites and festivals.
Respond: I am very grateful for your recommendations for related literature, which are very valuable and helped me comprehend the place of State Shinto in Imperial Japan. My modification is placed in endnote6, and explain why I prefer to use the term “quasi-religion”:
“It is noteworthy that the applicability of Shinto to the concept of “religion” has been controversial since it was introduced to Japan along with a number of Euro-American concepts in the 1870s (Josephson 2012, 94). The bureaucrats in the Meiji era strove to draw a clear boundary between the secular and religious spheres, avoiding to define the State Shinto as a religion and especially a state religion, but rather the rites of state, patriotic morality in which all people were compelled to partici-pate. The distinction between State Shinto and other religions such as Buddhism, Christianity and Sect Shinto was eventually confirmed in the 1889 Imperial Constitution, as Trent E. Maxey commented “the Constitution codified the religious settlement by explicitly rejecting religion as a component of national definition. It thus adopted the principle of religious freedom over toleration (Maxey 2014, 14).” This constituted what Yasumaru Yoshio calls the “Separation of church and state of Japanese type”, in which State Shinto was rites of state in public sphere requiring mandatory participation, while the religious affairs were restricted to the private sphere, and individuals had the constitutional right of the freedom of belief (Yasumaru 1979, 208-09). However, it cannot be ignored that the distinction was more confined to the legal and administrative level. State Shinto contained many religious elements, from historical soureces and mythology to the ritual with its temporal and spatial dimensions. In reality there were still multiple cases of conflicts between State Shinto and religious beliefs, especially in the area of individual spirituality, such as the lèse-majesté incident of Uchimura Kanzo and the 1932 Sophia University— Yasukuni Shrine incident. Therefore, this article prefers to define State Shinto as “quasi-religion” and discusses its rhetoric with religious dimension.”
p.5: “the GHQ” – give us the full name the first time it is used, and maybe contextualize it. Not everyone is familiar with how the occupation was organized.
Respond: I accepted your comment, and added the full name of GHQ in the text.
p.5: “Nation-Body” – Kokutai is very commonly translated as “national polity” in English. Not saying the author has to follow this pattern, but it is well established.
Respond: Thank you for your reminder, and I have revised my expression to “Nation Polity”.
p.7: The narrative presented here about Tanaka’s wish to make Nichirenism a “state religion” in Japan would really benefit from a better contextualization, in particular with regards to “religion” under the 1890 constitution. I really think the author should look at Maxey’s work (see above).
Respond: I am grateful for your comment and recommendation. In the historical context of Imperial Japan, there was actually no state religion. According to the 1889 Constituion, freedom of religion and seprartion of church and state were also guaranteed under certain conditions. In Tanaka’s project, Nichirenism incorporated some elements of the mythology of Kokutai as well as veneration of emperor. His project was precisely to change the constitutional principle of separation of church and state so that Nichirenism would be established as the state religion.
p.8: “As scholar Yulia Burenina commented” – fairly superfluous use of “scholar”, the reader assumes that those cited are scholars, unless otherwise noted.
Respond: I deleted the expression “scholar”.
p.11: “far from the animistic Shinto with belief in kami(spirits)” – referring to Shinto as “animist” is stepping into an unnecessary and deeply polemic discussion about the use of “animism” in the study of religion in Asia. There is no need to do so here, so I would just avoid the term. Also, the discussion in this paragraph would again benefit from a better understanding of the constitutional debates that resulted in Shinto being placed outside of the discourse of “religion” in Meiji Japan. Maxey’s work is really good here, but perhaps the author should also consider Jason Josephson Storm’s “The Invention of Religion in Japan” (2012), as a way of deepening the discussion.
Respond: Thank you for your comment, and I likewise find it inappropriate to identify Shinto as animism, and irresponsible to make such an analogy with State Shinto. Shinto also had a variety of categories that could not be summarized simply as animism prior to Meiji era. I think it’s understandable of the status of State Shinto under 1889 Constitution and how it related to religion such as Buddhism, Christianity and Sect Shinto. I accepted your comment, and decided to delete the comparision.
p.11: “Scholar Purdy argued…” – Again, unnecessary use of “scholar” here.
Respond: I deleted the expression “scholar”.
p.12: “When living in the Japanese Empire, one would orient oneself in time according to three types of chronology in parallel: firstly, the Common Era chronology…” – this is not correct, right? While the common era was used by some (in particular Christians) prior to 1945, the reign era system would have been fully dominant in Imperial Japan.
Respond: I found that the Common Era chronology was indeed very rare in Imperial Japan. This was a mistake in my paper, so I decided to delete it and added endnote14:
“Although the Anno Domini was introduced to Japan as early as the Meiji era, it was not as commonly used as in other Asian countries prior to 1945. And it is often translated as Seireiki 西暦 (Western Year).”
p.13: The project of building the tower using resources from the whole empire as described here reminds me of how the Great Buddhas in Nara and Kamakura were funded and built. Was there a conscious parallel to these more ancient national projects as well?
Respond: I think this is a very interesting suggestion. Unfortunately, limited to the materials in my possession, I have yet to find an exact connect between the Hakkō Ichiu Tower in Miyazaki with the Great Buddhas in Nara or Kamakura. But it’s still very enlightening and hopefully this connection will be found in my subsequent studies.
p.15: “Tanaka was not a follower of State Shinto but a firm nationalist” – this sentence is incomprehensible. State Shinto, if that is what the author wants to call the “secular” system of mandatory shrine rites and emperor worship prior to 1945, was not something you could “choose” to follow. All imperial subjects were part of this system, just as they were part of the Empire of Japan. Religion had followers, “State Shinto” did not. It only had imperial subjects. Again, I think Maxey’s work would be useful here.
p.15 (cont.): The discussion about “syncretism” is also a bit off. Buddhism and mandatory shrine rites coexisted throughout most of the Meiji period, and through all of Taisho and imperial Showa. Under the 1890 constitution Buddhism was simply compartmentalized in its own sphere as a “religion” and was therefore optional, whereas shrines were not. Tanaka didn’t question the emperor’s divinity, because to do so would be lèse-majesté. He wanted a place for Buddhism in the mandatory national narrative, he did not “reinvent” syncretism. Again, Maxey discusses exactly these debates.
Respond: Thank you again for helping to clarify the relation between State Shinto and religions. I accepted your comment, and also think it’s irresponsible to regard “Tanaka was not a followe of State Shinto.” So, I deleted it. And for the “syncretism” issue, I modified the expression:
“Although State Shinto was not a religion in the historical context, Tanaka still fostered Nichirenism to be one of the most important nationalist forces in Imperial Japan, by incorporating the elements from State Shinto and Kokutai.”
I think that the religious circumstance in Imperial Japan was more complex than what I have expressed in the article. Many religions, including Nichirenism and Ōmoto, were actually influenced by the mythology of State Shinto, and incorporated some elements of the Kokutai doctrine into their own narratives, which was more activite before the enactment of Peace Preservation Law in 1925. (The Peace Preservation Law explicitly prohibited altering Kokutai.) These religious followers became important forces in the nationalist movement of the Showa era.

Reviewer 3 Report
This article gave an overview of the important wartime phrase "Hakko Ichiu", starting with its etymological and figurative meaning in Chinese and then how it was rediscovered by the founder of Nichirenism, who envisioned a world-wide government that had Nichiren Buddhism as a state religion. Later, the term was widely promoted by the Japanese state in order to legitimize the Second Sino-Japanese War, but shifted in meaning to refer to creating a Greater East Asia Co-Prosperity Sphere not directly ruled by Japan, but with Japan at its core and all nation's embracing a vague "spirit of Jinmu". Although the phrase Hakko Ichiu appears in many academic works concerning Japanese imperialism, its development is often glossed over. This paper provides an important contribution to research by delving into the evolution of this vital phrase, and will serve as a useful citation for those wishing to direct readers somewhere to learn more about the term. However, I feel some points in the paper need to be clarified.
p5: It is unclear on how the Shogunate took note of Shinto "spontaneously". Please clarify.
p8: "...in the official newspaper Kokuchuu Shinbun". Is this the official newspaper of the Nichiren sect? Of Nichirenism specifically?
p10: A little bit more about what "Jinmu's spirit" referred to during this time period would be helpful, even if just saying that it was a term whose meaning was left vague.
p11: Providing romanization for 行け八紘を宇となし would help explain the connection between Hakko Ichiu and this song for people who don't read Chinese characters. I recommend incorporating note 12 into the body of the text.
p11: Also, the citation of "AussieMinecrafter" is inappropriate for this translation, as he clearly does not speak Japanese and merely scrapped the translation from somewhere else on the internet. Rather, isn't this the official translation of the song by the prewar Foreign Ministry's Obata Shigeyoshi 小畑薫良 (1888-1971)?
p11: I find it problematic to make the jump from "something similar to Christianity" to a "state religion". There are now quite a few works complicating this picture of the Meiji government's relationship with Shinto. May I recommend Trent Maxey's The Greatest Problem, Jason Josephson's The Invention of Religion in Japan (both on Shinto's relationship to the category of religion), and Tomoko Masazawa's The Invention of World Religions (on the definition of religion in the West)? I think contrasting a Christianity-like State Shinto with "animistic Shinto" needs more justification before it in included in this paper.
p13: "Miyazaki...is the birthplace of Emperor Jinmu". It might be best to reword this in order to avoid implying Emperor Jinmu is an uncontested historical figure.
p13: "Emperor Jinmu's last palace". The last palace Emperor Jinmu ever built? Is this the same as the one he built on Kigensetsu? It might be helpful to clarify.
p15/16: Again, this was a fascinating discussion about the shift from Nichirenism to the wartime rhetoric. I wonder if the difference between the older Nichirenism meaning and the wartime meaning could be highlighted more in your conclusion? Also, expanding your discussion beyond the Miyazaki tower to mention the situation overseas (for example, the Hakko Ichiu tower at Chosen Jingu in Korea and the use of the phrase in Manchurian migration) could make this paper even more useful. I've also seen the translation of Hakko ichiu as "universal brotherhood" used in relation to Manchuria, so discussing that briefly would also be a great addition.
There were also several typos within the article. Please have an editor go through the paper closely again. For example:
P5: "The Marth 13 Decree" (should be "March 13")
p7: "Tanaka 1901a, 2" font is of a different size
p8: "The Scared Work of World Unification" (should be "The Sacred Work...")
p10: "which the young military officers wanted to be handed to the Emperor" is grammatically confusing and needs rewording
p11: "flock" and then "herd": this is a mixed metaphor (a flock of sheep and then a herd of cattle). Maybe sticking to one or the other might be better?
p12: Common Era (CE) is used, but then BC (rather than BCE) is used, making the terminology inconsistent. Also, using "Common Era" is anachronistic, since the Japanese empire specifically didn't see the CE/AD chronology as "common" to Asia. I suggest using the historically accurate term (AD, not obscuring its connection to Christianity) or a translation of the Japanese term (西暦, Western Year).
p13: Sokoku 祖国 should be translated as "ancestral land" not "homeland".
p14: "Yasuhito Prince Chichibu" should be "Prince Yasuhito Chichibu"
p14: I wonder if "braziers" might be a better term than "bonfire pavilions", based on the picture. Pavilion usually refers to something with at least a roof.
Also, the quote on p6: "How felicity we are born..." Could this perhaps be a typo for "How felicitous..." or "What felicity..."?
Author Response
Reply to Comments on
Manuscript religions-2070449:
“Hakkō Ichiu: Religious Rhetoric in Imperial Japan”
Dear Editors and Reviewers:
The revised version of the manuscript (Manuscript religions‑2070449) has been submitted, which has been cautiously revised according to the reviewers’ comments.
I would like to express my sincere gratitude to Editors and three anonymous reviewers for their precious time and constructive suggestions. Their insightful comments are very helpful in improving the quality and presentation of this paper. All the comments have been seriously considered and addressed in the revised manuscript.
Below is an item-by-item response to the comments from three anonymous reviewers.
Reviewer 3’s Comments
This article gave an overview of the important wartime phrase "Hakko Ichiu", starting with its etymological and figurative meaning in Chinese and then how it was rediscovered by the founder of Nichirenism, who envisioned a world-wide government that had Nichiren Buddhism as a state religion. Later, the term was widely promoted by the Japanese state in order to legitimize the Second Sino-Japanese War, but shifted in meaning to refer to creating a Greater East Asia Co-Prosperity Sphere not directly ruled by Japan, but with Japan at its core and all nation's embracing a vague "spirit of Jinmu". Although the phrase Hakko Ichiu appears in many academic works concerning Japanese imperialism, its development is often glossed over. This paper provides an important contribution to research by delving into the evolution of this vital phrase, and will serve as a useful citation for those wishing to direct readers somewhere to learn more about the term. However, I feel some points in the paper need to be clarified.
Allow me to express my sincerest gratitude for your valuable comments, which are extraordinary important both for enhancing this article and for my future research. My itemized responses are below.
p5: It is unclear on how the Shogunate took note of Shinto "spontaneously". Please clarify.
Respond: It is a complex issue how the Meiji government took note of Shinto. I expressed it too briefly. Maybe it’s better to mention the sources of Fukkō Shintō from Hirata Atsutane :
“Although the Meiji government, before the creation of parliament in 1890, was de facto an oligarchy, it was ostensibly an absolute monarchy ruled by the emperor. In consideration of the elements of veneration of the emperor and the idea of Kokutai from Fukkō Shintō 復古神道 advocated by Hirata Atsutane 平田篤胤 in the Edo period (Hardacre 2017, 348), the government spontaneously took note of Shinto.”
p8: "...in the official newspaper Kokuchuu Shinbun". Is this the official newspaper of the Nichiren sect? Of Nichirenism specifically?
Respond: It is the official newspaper of Nichirenism. “In 1913, Tanaka published an essay called Emperor Jimmu’s National Foundation on Kokuchū Shinbun『国柱新聞』, the official newspaper of Risshō Ankokukai.”
p10: A little bit more about what "Jinmu's spirit" referred to during this time period would be helpful, even if just saying that it was a term whose meaning was left vague.
Respond: I added the following:
“It was often articulated as Jimmu’s entrepreneurship spirit (神武創業の精神) or spirit of national foundation(肇国の精神). During the Meiji Restoration period, the promotion of Jimmu’s spirit was often associated with the abolition of the shogunate system and direct imperial rule, but its exact meaning was still left vague.”
p11: Providing romanization for 行け八紘を宇となし would help explain the connection between Hakko Ichiu and this song for people who don't read Chinese characters. I recommend incorporating note 12 into the body of the text.
Respond: I accepted your comment, adding the romanization of this sentence, and put the endnote into the text.:
“Let us make the world our home, call to fellow men (Yuke Hakkō wo Ie to nashi 往け八紘を宇となし).”
“The six line (the fifth line in Japanese version) is literally translated as “Carry up the eight cords (Hakkō) to be the roof,” and it is a variant of Hakkō Ichiu, expressing the ideal of universal brotherhood under the emperor.”
p11: Also, the citation of "AussieMinecrafter" is inappropriate for this translation, as he clearly does not speak Japanese and merely scrapped the translation from somewhere else on the internet. Rather, isn't this the official translation of the song by the prewar Foreign Ministry's Obata Shigeyoshi 小畑薫良 (1888-1971)?
Respond: I am very grateful for giving me the clue of this translation. I found the source of the official translation of this song, according your precious clue.
“(Obata Shigeyoshi 1938) Obata Shigeyoshi 小畑薫良. Trans. 1938. “Patoriotic March,” The Rising Generation 英語青年 Vol. 79. No.1: 27”
p11: I find it problematic to make the jump from "something similar to Christianity" to a "state religion". There are now quite a few works complicating this picture of the Meiji government's relationship with Shinto. May I recommend Trent Maxey's The Greatest Problem, Jason Josephson's The Invention of Religion in Japan (both on Shinto's relationship to the category of religion), and Tomoko Masazawa's The Invention of World Religions (on the definition of religion in the West)? I think contrasting a Christianity-like State Shinto with "animistic Shinto" needs more justification before it in included in this paper.
Respond: Thank you for your comment and recommendations. I likewise find it inappropriate to identify Shinto as animism, and irresponsible to make such an analogy with State Shinto. Shinto also had a variety of categories that could not be summarized simply as animism prior to Meiji era. I think it’s understandable of the status of State Shinto under 1889 Constitution and how it related to religion such as Buddhism, Christianity and Sect Shinto. I accepted your comment, and decided to delete the comparision.
Meanwhile I added endnote five to clarify the relation between State Shinto and religions, for replying another reviewer’s comment, and I think it is also related to this issue, so please allow me to put the endnote6 here in response.
“It is noteworthy that the applicability of Shinto to the concept of “religion” has been controversial since it was introduced to Japan along with a number of Euro-American concepts in the 1870s (Josephson 2012, 94). The bureaucrats in the Meiji era strove to draw a clear boundary between the secular and religious spheres, avoiding to define the State Shinto as a religion and especially a state religion, but rather the rites of state, patriotic morality in which all people were compelled to partici-pate. The distinction between State Shinto and other religions such as Buddhism, Christianity and Sect Shinto was eventually confirmed in the 1889 Imperial Constitution, as Trent E. Maxey commented “the Constitution codified the religious settlement by explicitly rejecting religion as a component of national definition. It thus adopted the principle of religious freedom over toleration (Maxey 2014, 14).” This constituted what Yasumaru Yoshio calls the “Separation of church and state of Japanese type”, in which State Shinto was rites of state in public sphere requiring mandatory participation, while the religious affairs were restricted to the private sphere, and individuals had the constitutional right of the freedom of belief (Yasumaru 1979, 208-09). However, it cannot be ignored that the distinction was more confined to the legal and administrative level. State Shinto contained many religious elements, from historical soureces and mythology to the ritual with its temporal and spatial dimensions. In reality there were still multiple cases of conflicts between State Shinto and religious beliefs, especially in the area of individual spirituality, such as the lèse-majesté incident of Uchimura Kanzo and the 1932 Sophia University— Yasukuni Shrine incident. Therefore, this article prefers to define State Shinto as “quasi-religion” and discusses its rhetoric with religious dimension.”
p.13: "Miyazaki...is the birthplace of Emperor Jinmu". It might be best to reword this in order to avoid implying Emperor Jinmu is an uncontested historical figure.
Respond: Thank you for your comment. I have made the following changes as I did not express it accurately here.
“As mentioned above, Miyazaki Prefecture, formerly known as Hyūga, was believed as the birthplace of Emperor Jimmu and the origin of the sacred imperial state of Japan in State Shinto mythology”
p13: "Emperor Jinmu's last palace". The last palace Emperor Jinmu ever built? Is this the same as the one he built on Kigensetsu? It might be helpful to clarify.
Respond: Thank you again for pointing out the mistake. There was some confusion in my previous expressions, and the palace in Miyazaki was obviously not Jimmu’s last palace. I changed it as following:
He then laid out the basics of his plan “We will select a suitable site in the site of Emperor Jimmu’s palace prior to eastern expedition, and erect the magnificent Pillar of Heaven in the purely Japanese style, made of solid stones (Hakkō 2017, 82).”
p15/16: Again, this was a fascinating discussion about the shift from Nichirenism to the wartime rhetoric. I wonder if the difference between the older Nichirenism meaning and the wartime meaning could be highlighted more in your conclusion? Also, expanding your discussion beyond the Miyazaki tower to mention the situation overseas (for example, the Hakko Ichiu tower at Chosen Jingu in Korea and the use of the phrase in Manchurian migration) could make this paper even more useful. I've also seen the translation of Hakko ichiu as "universal brotherhood" used in relation to Manchuria, so discussing that briefly would also be a great addition.
Respond: I am very grateful for your suggestions, which would enhance my conclusion. Hakkō Ichiu had escaped from the context of Nichirenism in the late 1930s and turned into a more widely used propaganda slogan. In Tanaka’s project, Hakkō Ichiu represented the grand ideal of proselytizing Nichirenism and achieving world unification through moral and missionary means. But the later Nichirenists in 1930s preferred to resort violent means to achieve this ideal. Ishihara Kanji had conceived the World Final Armageddon, but I don’t think he got this idea from Tanaka alone. I add below in my conclusion:
“Although State Shinto was not a religion in the historical context, Tanaka still fostered Nichirenism to be one of the most important nationalist forces in Imperial Japan, by in-corporating the elements from State Shinto and Kokutai. Furthermore, Nichirenism pro-foundly influenced the nationalist movement in the early Showa era, and many military officers, including Ishiwara Kanji, were followers of Nichirenism, whose radical actions accelerated the Imperial Japan to disastrous wars.”
For the second point, according to my possession, there were many stone monuments, stone pillars and stone lanterns in shrines throughout Japan carved with motto Hakkō Ichiu. This was particularly prevalent in Hiroshima Prefecture, probably because a major base of the Imperial Navy was located in Kure, Hiroshima. I believe that there were many similar relics in overseas shrines, but unfortunately, I have not seen the picture information of this aspect till now. I add below in my conclusion:
“In addition to the Hakkō Ichiu Tower, the motto was often carved on stone monuments and pillars of shrines throughout Japan and overseas, manifesting the ideal of “universal brotherhood”.”
There were also several typos within the article. Please have an editor go through the paper closely again. For example:
P5: "The Marth 13 Decree" (should be "March 13")
Respond: I have revised it.
p7: "Tanaka 1901a, 2" font is of a different size
Respond: I have revised it.
p8: "The Scared Work of World Unification" (should be "The Sacred Work...")
Respond: I have revised it.
p10: "which the young military officers wanted to be handed to the Emperor" is grammatically confusing and needs rewording
Respond: I changed my expression as following:
“which the young military officers wished to submit to the Emperor in the February 26 Incident in 1936, advocated,”
p11: "flock" and then "herd": this is a mixed metaphor (a flock of sheep and then a herd of cattle). Maybe sticking to one or the other might be better?
Respond: I unified the two expressions into one word “flock”.
p12: Common Era (CE) is used, but then BC (rather than BCE) is used, making the terminology inconsistent. Also, using "Common Era" is anachronistic, since the Japanese empire specifically didn't see the CE/AD chronology as "common" to Asia. I suggest using the historically accurate term (AD, not obscuring its connection to Christianity) or a translation of the Japanese term (西暦, Western Year).
Respond: I found that the Common Era chronology was indeed very rare in Imperial Japan. This was a mistake in my paper, so I decided to delete it and added endnote14:
“Although the Anno Domini was introduced to Japan as early as the Meiji era, it was not as commonly used as in other Asian countries prior to 1945. And it is often translated as Seireiki 西暦 (Western Year).”
p13: Sokoku 祖国 should be translated as "ancestral land" not "homeland".
Respond: I modified it to “ancestral land”.
p14: "Yasuhito Prince Chichibu" should be "Prince Yasuhito Chichibu".
Respond: I modified it to “Prince Yasuhito Chichibu”, and changed “Prince Takamatsu Nobuhito”to ‘Prince Nobuhito Takamatsu”.
p14: I wonder if "braziers" might be a better term than "bonfire pavilions", based on the picture. Pavilion usually refers to something with at least a roof.
Respond: I modified the term to “braziers”.
Also, the quote on p6: "How felicity we are born..." Could this perhaps be a typo for "How felicitous..." or "What felicity..."?
Respond: I changed it to “How fortunate that…”

Round 2
Reviewer 3 Report
The author has addressed all the concerns I raised, and I look forward to the publication of the article. The newly added sections have a few minor grammatical errors, so I hope an editor will read over those sections closely.